# Maternal skepticism regarding children's vaccination in Jordan: Investigating the knowledge, attitude, and adherence

**Rand A. Al-Rashdan**[1]**, Wafa'a Ta'an**[2]***, Tareq Mukattash**[3]**, Brett Williams**[4]

**1** Faculty of Medicine, Jordan University of Science and Technology, Irbid, Jordan, **2** Faculty of Nursing, Jordan University of Science and Technology, Irbid, Jordan, **3** Faculty of Pharmacy, Jordan University of Science and Technology, Irbid, Jordan, **4** Department of Paramedicine, Monash University, Clayton, Victoria, Australia

* wftaan@just.edu.jo

## Abstract

Vaccination is critical to every responsible healthcare system, yielding several health and economic benefits. However, skepticism is a major challenge to vaccination adherence among parents globally. Mothers are primary decision-makers on children's vaccination; therefore, this study aims to assess maternal knowledge, attitudes, and adherence to children's routine vaccination in Jordan, identify the main sources of related information, and explore the awareness and hesitancy related to elective vaccination, which is not incorporated into the national vaccination program (e.g., varicella, flu, meningitis, and meningococcal vaccines). A descriptive, cross-sectional design was conducted to collect data from 533 Jordanian mothers between August and November 2023. Participant mothers completed an online self-administered questionnaire comprising demographics, knowledge, attitudes, and adherence to children's vaccination. The surveyed mothers in general exhibited high levels of knowledge, attitudes, and adherence concerning children's vaccination and a relatively moderate level of adherence concerning elective vaccines. The mothers' perspectives of reluctance towards elective vaccines were explored. The primary healthcare center and physicians were the most prevalent sources of mothers' vaccine-related information. Future efforts should implement tailored health education campaigns that address specific knowledge gaps, such as misconceptions about contraindications and side effects, promote awareness about elective vaccines, and engage healthcare providers to facilitate informed decision-making among Jordanian mothers and improve children's vaccination coverage.

**Data Availability Statement:** All relevant data are within the manuscript.

**Funding:** The author(s) received no specific funding for this work.

## Introduction

Vaccination programs are considered essential components of responsible healthcare systems, and several health and economic advantages result from effective vaccination practices [1]. Various research studies have acknowledged that vaccination programs are cost-effective approaches to reducing disease-specific morbidity and mortality in a population leading to an improved overall quality of health [e.g., 2]. The World Health Organization [3] published a

**Competing interests:** The authors have declared that no competing interests exist.

report stating that vaccinations among children may reduce an estimated 3.5–5 million deaths per year from vaccine-preventable diseases (VPDs) such as diphtheria, tetanus, pertussis, influenza, and measles. Studies also showed that widespread availability and adoption of vaccinations have substantially reduced morbidity, mortality, and healthcare expenses related to infectious illnesses and enhanced the quality of life for people worldwide [4].

## Impact of vaccination campaigns

Vaccination programs have significantly led to a reduction in the prevalence of VPDs as well as the associated complications [5]. Overall, vaccines are known to reduce the possibility of contracting certain infectious diseases, such as polio, and have been utilized for decades in many countries around the world [6]. The most prevalent VPDs include rubella, measles, diphtheria, tetanus, pertussis, and polio. Literature shows that more than three million lives each year can be saved by vaccinations across the world. For example, the polio vaccine has prevented the paralysis of an estimated 5 million individuals who otherwise would have contracted the disease [7].

## Jordanian context

Jordan is a low-middle-income country located in the Middle East [8]. It has received over 750,000 refugees, which makes it one of the world's largest refugee populations residing in Jordan, due to wars in nearby countries that affected the economic ability of the country [9]. Consequently, the vaccination program in Jordan has been affected by the high pressure placed on the health system [10]. The United Nations Children's Fund (UNICEF)- Jordan supports the Jordanian Ministry of Health's vaccination program to ensure access to remote locations and marginalized populations, including non-Jordanian children [11]. UNICEF collaborates with the government of Jordan to strengthen the national children and youth health promotion programs by facilitating the supply of vaccinations, medical equipment, and technical experience. This collaboration offers a free children's vaccination program including Bacille Calmette Guerin (BCG) vaccine, Hexaxim vaccine (Hexa), Inactivated Poliovirus vaccine (IPV), Diphtheria, Tetanus, and Pertussis vaccine (DaPt), Haemophilus influenza vaccine, type B (Hib), ROTA vaccine, Oral Polio Vaccine (OPV), and Measles, Mumps, and Rubella vaccine (MMR) [11]. This program has been within Jordanian laws since 1979. The vaccination in Jordan has helped reduce fatalities and outbreaks of avoidable diseases [11].

## Elective vaccines

Every country has developed its immunization schedule, which is provided free of charge beginning at birth, based on socioeconomic and epidemiological factors. A variety of elective vaccines that can be purchased on an individual basis are also available. The latter immunizations are voluntary and are neither rewarded nor reimbursed [12]. The use of elective vaccines is influenced by several socioeconomic factors as well as the doctors' engagement in educating parents about alternative prophylactic choices [13, 14]. Elective vaccines available in Jordan include; Pneumococcal, Flu, Varicella, and Meningococcal vaccines [15, 16]. Interestingly, some of these elective vaccines are included in the mandatory vaccination programs in some countries like USA and Canada. However, no studies are addressing parental awareness towards elective vaccines in Jordan.

## Challenges to successful vaccination

Despite that there are children's vaccination programs in place, there have been some challenges to the successful implementation of vaccination programs as evidenced by fluctuations in the numbers of VPD cases in many countries, notably in the Middle East region [17]. The World Health Organization [18] indicated that about 20 million did not receive vaccines against three diseases; including measles, diphtheria, and tetanus, and a large number of children have died from these diseases. The number of death cases reported for VPDs can be linked to the skepticism and hesitation among mothers to take their children for immunization. Additionally, Batarseh et al. performed computer-assisted personal interviews to assess the perception and attitude of the public on vaccines' current practices in Jordan and register an average child vaccination. The study concluded that there is a pressing need to enhance public education and enhance vaccination facilities and personnel [19]. A recent study in Jordan, revealed an elevated negative attitude regarding children's vaccination post the COVID-19 pandemic, indicating hesitancy and diminished trust related to the safety and efficacy of the vaccines [20].

## Parental role

Parental choices on vaccination are critical for increasing vaccination rates and compliance. Therefore, the knowledge, attitude, and adherence of parents toward vaccination need to be assessed to evaluate parents' vaccination choices [21]. Parents', and more specifically mothers', cooperation is proven to enhance the successful completion of children's vaccinations and reduce the likelihood of vaccine failures [2]. For example, a study conducted in Greece underlined vaccine hesitancy among pregnant women in Athens. The study indicated a below-average intention to vaccinate children. About 10% of women were unable to decide whether to vaccinate their children; the contributing factors included inadequate information from the physicians concerning vaccine safety [22]. Previous studies investigated the factors affecting parents' vaccination choices, including various socioeconomic factors as well as the information provided to parents about available prophylaxis options [13, 14]. Statistically, a lack of knowledge contributes to poor vaccination practice [23]. Matta et al. underlined that the contributing factors include maternal sociodemographic factors, maternal trust in the public health systems, the relationship between mothers and physicians, mothers' level of knowledge, and their attitudes towards vaccinations. Better vaccination practice among mothers depends on mother-physician communication, which promotes better knowledge and elevates maternal trust in vaccines [17].

## COVID effect

It is also important to mention that due to the COVID-19 epidemic, there has been a significant decline in children's vaccination coverage globally, and in Jordan specifically [20]. The results of cross-sectional studies showed that adherence to childhood vaccination programs in Jordan was a challenge for the year 2020, with a substantial decline in vaccination coverage. This study also indicated that Jordan's health system was under significant pressure during the pandemic due to the economic situation as well as the large influx of Syrian refugees that resulted in a decline in vaccination coverage. However, it was challenging to accurately estimate the vaccination coverage for non-Jordanians since there was insufficient data available [24].

In summary, numerous studies have investigated parental knowledge, attitude, and adherence to children's vaccinations worldwide, providing diverse outcomes impacted by factors unique to each community and country [e.g.: 2, 5, 12, 17, 21]. However, variations existed in

the results of the previous studies. For example, studies in South Africa showed a low level of knowledge but a positive attitude and practice [2]. Also, limited knowledge was uncovered by a study in Lithuania [5]. Weak knowledge, negative attitude, and practice were found in a study in Lebanon [17]. On the other hand, a study in Saudi Arabia found good knowledge, positive attitude, and good adherence [21]. This is consistent with the findings of Almutairi et al. [25]. In addition, there is a scarcity of published research investigating the progress in the level of mothers' knowledge, attitude, and adherence to children's vaccination in Jordan, Therefore, further studies were recommended to investigate the knowledge, attitude, and adherence to vaccinations within their specific contexts.

This descriptive study aims to assess Jordanian mothers' knowledge, attitudes, and adherence to children's vaccination. Additionally, it assesses mothers' willingness to provide their children with elective vaccines and the contributing factors to declining elective vaccines. The sources of medical information will also be assessed to offer stakeholders insights to establish strategies that maximize mothers' adherence to children's vaccination programs. A secondary aim was to assess perceptions related to pharmacists' roles in vaccination. This study will contribute to the body of knowledge by providing a thorough understanding of maternal knowledge, attitude, and adherence to children's vaccination to guide national efforts and strategies aiming to enhance vaccination rates.

## Materials and methods

### Study design

The study adopted a cross-sectional descriptive design supported by quantitative data from mothers' perspectives to gauge their knowledge, attitudes, and adherence to children's vaccination. The selection of a cross-sectional design was favorable, primarily due to its capacity to mitigate the impact of spatial and temporal factors through the administration of the study instrument within a specific time frame across the entire sample [26].

### Setting and sampling

A convenience sampling technique was employed to collect pertinent data concerning the study's objectives. The selection of appropriate sample size was based on the Cochrane formula, a method renowned for its usefulness, particularly within scenarios involving large population sizes [27]. This formula is embodied as follows $n_0 = \frac{Z^2 pq}{e^2}$ where: $n_0$: sample size in unlimited populations, $Z^2$: square of the confidence level in standard error units (Z = 1.96 at 95% confidence interval level), $p$: estimated proportion of successes (p = 0.50), $q$: estimated proportion of failures (1-p = 0.50), $e^2$: square of the maximum allowance for error between the true proportion and the sample proportion (e = 0.05). Accordingly, the appropriate sample size estimated in the study after applying the previous formula was as follows: $n_0 = \frac{(1.96)^2(0.50)(0.50)}{(0.05)^2} \cong 385$.

The study questionnaire was distributed online through prevalent social media platforms such as Facebook, Instagram, and WhatsApp, strategically aiming to achieve an optimal response rate. The eligibility criteria included mothers aged 18 years or higher who demonstrated voluntary participation in the study, possessed at minimum one child below the age of 12 and were residents within the confines of Jordan. The final sample in this study was composed of 533 responses. Remarkably, the aggregated sample size covered the designated threshold of minimal sampling adequacy. This significant expansion confirms the fulfillment of the predetermined objective of sampling sufficiency [28].

## Study instruments

The study employed a self-administered questionnaire that includes the socio-demographic questionnaire and questions adapted from previous literature done in the English language [e.g., 25]. To improve comprehension among respondents, all items were translated into Arabic. Subsequently, these translated items were reconverted to English using a reverse translation process to ensure their content authenticity as advocated by Lee et al. [29]. However, each item was provided in both Arabic and English format.

The final version of the questionnaire included two sections. The first section was dedicated to collecting demographic data, including mothers' age, location of residence, educational level, job status, working/studying in a medical field, income level, number of children, primary sources of information, and ability to obtain knowledge and information to manage health concerns (ability to access health information). The second section of the questionnaire is derived from the research conducted by Almutairi et al. [25], with the addition of four new items aimed at evaluating attitudes, knowledge, and adherence to elective vaccination in Jordan. Additionally, an inquiry about respondents' perceptions toward pharmacists' roles in the vaccine sector was added.

The second section comprises a comprehensive assemblage of 24 items. The first 12 items assessed the mothers' knowledge of children's vaccinations. This section is supplemented by a query demonstrating the sources of information the respondents used concerning vaccinations. The mothers could convey their perspectives on the knowledge items by providing a response of "yes" or "no". The total knowledge score is determined by adding up the values of their responses, which can range from 12 to 24. Higher scores signify greater knowledge. The next six items measured maternal attitudes towards children's vaccinations. The responses to these items were designed through a five-point Likert scale, including strongly disagree (1), disagree (2), neutral (3), agree (4), and strongly agree (5). The overall attitude score is computed by summing the values of their responses, which can fall within a range of 5 to 30. Elevated scores indicate a more favorable attitude.

Another six items measured maternal adherence to children's vaccinations. Respondents were asked to rate the items in this section on a three-point Likert scale, including always (3), sometimes (2), and never (1). The total adherence score is calculated by summing the values of their responses, which can vary between 6 and 18. Higher scores indicate a more consistent adherence. The last item of the adherence section was "I give my child elective vaccines (which are not included in the national vaccination program)". The participating mother was asked to complete the next prompt if her answer was never which is a multiple-choice item to identify the reason for not participating in elective vaccinations with the ability to select more than one option.

## Data collection

The questionnaire was designed in an electronic format using Google Forms and systematically published every week during the period between August 7th and November 7th, 2023. The introductory component of the questionnaire included a cover letter outlining the primary goals of the study, which was set in the context of Jordan's children's vaccination. Additionally, it included a clear confirmation of the researcher's commitment to confidentiality. The questionnaire was structurally designed with a pair of obligatory queries to ensure adherence to the prerequisites for engagement in the study. The initial query sought the respondent's confirmation that their age surpassed the threshold of 18 years and that they had at least one child at the age of 12 years or less. The second query focused on the agreement of voluntary approval, underpinned by the option of terminating the response at one's discretion.

After completing the questionnaire, the mothers were given the option to provide an email address to which they could receive information regarding children's vaccinations. After the data collection process ended, mothers received a link that directed them to the Ministry of Health's extensive vaccination booklet, which contains a summary of information on childhood vaccines within Jordan [30]. This authoritative manual elucidates essential details regarding both mandatory and elective vaccines accessible in Jordan, accurately explains their respective significance, contraindications, instances calling for the postponement of vaccination, and a vast collection of information relevant to the vaccination domain. This was done to express the researchers' appreciation of their participation and to increase awareness towards children's vaccination.

## Validity and reliability

The reliability and validity of the questionnaire evaluating maternal knowledge, attitudes, and adherence to children's vaccinations have been thoroughly investigated in pertinent literature [17, 25, 31]. Nonetheless, the present study conducted several measures to ensure robust levels of validity and reliability, thereby enhancing the potential for generalizability of its findings. The face validity of the instrument was well ascertained by subjecting it to the investigation of a panel of healthcare academic experts in the field of pediatrics from Jordanian universities. This panel evaluated the linguistic formulation and contextual appropriateness of the items, their judgments, and recommendations incorporated to refine the final formulation of the questionnaire.

Furthermore, a pilot study encompassing 20 respondents was undertaken to validate the comprehensibility of the variables assessed by the items. It should be noted that the responses yielded from this pilot study were excluded from the eventual sample as recommended by Teresi et al. [32]. The internal consistency of the questionnaire was evaluated using Cronbach's alpha coefficients. The composite scale in this study produced a coefficient value of (0.77), demonstrating the instrument's reliability as it surpassed the stated threshold of 0.70, an essential criterion by the stipulations delineated by Louangrath and Sutanapong [33].

## Ethical consideration

The Jordan University of Science and Technology's Institutional Review Board (IRB) approved the study procedures (Research ID: 20230448). Participant mothers' confidentiality was preserved by using anonymous questionnaires. Participation in the study was voluntary and the participant mothers had the right to withdraw at any time, there are no consequences for declining. A letter of informed consent was included as the opening page of the online survey. The letter included a brief description of the study's purpose and an explanation that there are no foreseeable risks to participating. If the person agrees to participate, by clicking on the 'I agree to participate' option, they can proceed to the survey on the next page. This ensures that informed consent was collected from each participant before participating in the research.

## Data analysis procedures

The Statistical Package for the Social Sciences version 26 (SPSS) was used for the data analysis. Descriptive statistics including frequency, percentage, mean, and standard deviation were performed to achieve the study objectives. The descriptive statistics aimed to provide an overview of the study participating mothers as well as the variables of knowledge, attitude, and adherence to children's vaccination.

**Table 1. Description of demographic characteristics and main study variables (n = 533).**

| Variables | Categories | Frequencies | Percentages % |
|---|---|---|---|
| Location of Residence | Amman | 238 | 44.7 |
| | Irbid | 194 | 36.4 |
| | Zarqa | 44 | 8.3 |
| | Other | 57 | 10.7 |
| Educational level | Secondary or high school | 95 | 17.8 |
| | Diploma or Bachelor's | 363 | 68.1 |
| | Postgraduate | 75 | 14.1 |
| Job-status | Employed | 223 | 41.8 |
| | Unemployed | 310 | 58.2 |
| Working/studying in a medical field | Yes | 150 | 28.1 |
| | No | 383 | 71.9 |
| Income level | Less than 500 JD | 186 | 34.9 |
| | 500 –Less than 1000 JD | 214 | 40.2 |
| | 1000 JD or more | 133 | 24.9 |
| Number of children | 1 Child | 142 | 26.6 |
| | 2–3 Children | 274 | 51.4 |
| | More than 3 Children | 117 | 22.0 |
| **Variables** | **Minimum** | **Maximum** | **Mean** | **SD** |
| Mothers' age | 19 | 55 | 33.15 | 5.83 |
| Ability to access and understand health information | 1 | 10 | 7.62 | 2.03 |
| Total knowledge score | 13.00 | 24.00 | 20.90 | 1.98 |
| Total attitude score | 6.00 | 30.00 | 22.07 | 6.94 |
| Total adherence score | 11.00 | 18.00 | 14.89 | 1.13 |

## Results

The findings presented in Table 1 provide an extensive summary of the descriptive statistics of the demographics and main study variables. The results showed that the mothers' ages demonstrated a notable range encompassing 19 to 55 years, with a mean (M) age of (33.15) years and a corresponding standard deviation (SD) of (5.83). Responses to the ability to access health information variable ranged from 1 to 10, (M = 7.62, SD = 2.03). This distribution indicates the wide range of ability to access and understand health information scores among mothers who participated in this study. Mothers' educational levels were divided into three categories: diploma or bachelor's degree (68.1%) was obtained at the first rank, followed by secondary or high school (17.8%) at the second rank, then postgraduate (14.1%) at the third rank. The job status of the respondents showed that (41.8%) were employed, while (58.2%) were unemployed. Moreover, (28.1%) worked or studied in a medical field, while the remaining (71.9%) did not.

The mean scores of mothers' knowledge, attitude, and adherence to children's vaccinations were 20.9 (SD = 1.98), 22.07 (SD = 6.94), and 14.98 (SD = 1.13), respectively. See Table 1. The detailed illustrations of the frequencies and percentages of each response to each item are reported in the Tables 2–4. Items in these tables are sorted from highest to lowest scores.

Table 2 presents a comprehensive overview of mothers' knowledge about children's vaccinations. Each item was assessed by the frequencies and percentages of responses. These items were ranked from high to low according to the percentages of correct responses. To offer insights into the depth of maternal knowledge regarding children's vaccinations. Table 3 offers

**Table 2. Descriptive statistics for knowledge items (n = 533).**

| Items | Response Scale | |
|---|---|---|
| | Yes (%) | No (%) |
| Does even a healthy child need vaccination? | 517 (97.0) | 16 (3.0) |
| Is vaccination important for children from the first day of birth? | 506 (94.9) | 27 (5.1) |
| Can vaccination keep children healthy? | 505 (94.7) | 28 (5.3) |
| Are some vaccinations related to fever and pain? | 476 (89.3) | 57 (10.7) |
| Can childhood vaccinations control measles? | 439 (82.4) | 94 (17.6) |
| Can the hepatitis B virus be prevented by vaccination? | 432 (81.1) | 101 (18.9) |
| Can diphtheria, tetanus, and pertussis be controlled through vaccinations? | 418 (78.4) | 115 (21.6) |
| Does vaccination prevent infectious diseases? | 410 (76.9) | 123 (23.1) |
| Does vaccination reduce death and disability? | 385 (72.2) | 148 (27.8) |
| Can vaccinations cause cramps and rashes? | 249 (46.7) | 284 (53.3) |
| Are you aware of elective vaccines available in the private sector and not included in the national vaccination program? | 235 (44.1) | 298 (55.9) |
| Are malnutrition, low fever, and diarrhea not contraindications to vaccination? | 172 (32.3) | 361 (67.7) |

insights into mothers' attitudes regarding children's vaccinations. Regarding adherence, the results (Table 4) reveal the distribution of responses across adherence items, capturing the spectrum of adherence levels.

Moreover, the mothers who responded 'never' to the item 'I give my child elective vaccines (which are not included in the national vaccination program).' (n = 237) were asked about the reasons behind refraining from elective vaccination. Because there are cumulative reasons, this question is designed to be multi-response, including not knowing about it, high cost, fear of side effects, satisfaction with the national vaccination program, and not knowing where to find them. Table 5 presents the number of recurrences and the corresponding percentages for each identified reason.

The most common reason cited for non-involvement in elective vaccinations was satisfaction with the national vaccination program, accounting for 187 (78.9%) instances.

**Table 3. Descriptive statistics for attitude items (n = 533).**

| Items | Response Scale: frequency (percent) | | | | |
|---|---|---|---|---|---|
| | Strongly Agree | Agree | Neutral | Disagree | Strongly Disagree |
| I advise my relatives and family to vaccinate their children. | 216 (40.5) | 191 (35.8) | 45 (8.4) | 12 (2.3) | 69 (12.9) |
| Vaccinations are beneficial. | 204 (38.3) | 211 (39.6) | 38 (7.1) | 14 (2.6) | 66 (12.4) |
| Elective vaccines should be added to the national vaccination program list. | 202 (37.9) | 179 (33.6) | 65 (12.2) | 18 (3.4) | 69 (12.9) |
| I support the compulsory vaccination programs designed by the Ministry of Health. | 199 (37.3) | 189 (35.5) | 55 (10.3) | 23 (4.3) | 67 (12.6) |
| It is safe to have my child vaccinated. | 173 (32.5) | 226 (42.4) | 53 (9.9) | 13 (2.4) | 68 (12.8) |
| Pharmacies should provide and administer vaccines. | 64 (12.0) | 142 (26.6) | 107 (20.1) | 115 (21.6) | 105 (19.7) |

Table 4. Descriptive statistics for adherence items (n = 533).

| Items | Response Scale | | |
|---|---|---|---|
| | Always (%) | Sometimes (%) | Never (%) |
| My child received the mandatory childhood vaccines. | 523 (98.1) | 9 (1.7) | 1 (0.2) |
| I follow the compulsory vaccination programs listed in the vaccination schedule. | 520 (97.5) | 11 (2.1) | 2 (0.4) |
| I give my child the mandatory vaccines at its scheduled time. | 495 (92.8) | 34 (6.4) | 4 (0.8) |
| I don't give my child the vaccine if he/she has a fever. | 425 (79.7) | 82 (15.4) | 26 (4.9) |
| I give my child elective vaccines (which are not included in the national vaccination program). | 130 (24.4) | 166 (31.1) | 237 (44.5) |
| I give my child the vaccine if he/she has a cold or mild flu without fever. | 63 (11.8) | 124 (23.3) | 346 (64.9) |

Interestingly, not knowing about these elective vaccinations was reported as a reason for non-involvement among 172 mothers, contributing to around (72.57%) of the distribution. The lowest responses were noted on the response 'fear of side effects' among 76 mothers, making up approximately 28% of them.

On the other hand, mothers were asked to identify sources of information that contribute to their knowledge about children's vaccinations through a multi-response question. Available options included primary health care centers, the Ministry of Health, physicians, friends and relatives, the internet, and social media. The most prevalent sources of information for maternal knowledge were the primary healthcare center and physicians, accounting for 255 (26.5%), and 253 (26%) instances, respectively. The Ministry of Health was listed among 181 (18.6%) of mothers. Friends and relatives played a role in providing information for maternal knowledge in 160 (16.5%) instances. Lastly, the Internet and social media contributed to maternal knowledge in 123 (12.7%) instances.

## Discussion

The results demonstrated a high level of knowledge regarding children's vaccinations in Jordan which is consistent with the conclusions of Nassar et al. [34] which showed good knowledge among the parents by a mean percentage of 78%, and Abu-Rish et al. [35]. In contrary, the results of Matta et al. reviled that only 27% of 3500 caregivers in Lebanon had good knowledge [17]. Mothers' recognition of early vaccination's importance signifies understanding the possibility of early exposure to diseases and the need for timely protection. This emphasis aligns with medical principles that advocate for early obligatory immunization to establish active immunity against a range of diseases [36]. However, from a realistic evidence-based

Table 5. Reasons for non-involvement in elective vaccinations (n = 237).

| Reasons | Frequencies | Percentages |
|---|---|---|
| Satisfaction with the National Vaccination Program | 187 | 78.90 |
| Not knowing it | 172 | 72.57 |
| High cost | 92 | 38.82 |
| Not knowing where to find them | 87 | 36.71 |
| Fear of side effects | 76 | 28.27 |

Note: Percentages do not sum to 100% due to the possibility of respondents selecting multiple responses.

practice perspective, practitioners acknowledge vaccines' possible side effects, which are generally mild and temporary according to Maltezou et al. [22]. Such practice aims to encourage mothers' informed decision-making practices.

The overall high level of mothers' knowledge emphasizes the success of healthcare education initiatives in effectively communicating medical concepts surrounding vaccination. Their understanding of obligatory vaccination reflects an appreciation for the community-wide benefits, demonstrating an awareness of herd immunity principles as mentioned by Matta et al. [17]. While the awareness of elective vaccines remains moderate in proportion to another research conducted in India [37]. This represents an opportunity for medical professionals to further educate mothers about additional vaccination options. In the same context, mothers in Jordan rely on multiple sources of information to obtain their knowledge about children's vaccinations. Primary healthcare centers, physicians, and the Ministry of Health serve as primary sources. These sources, backed by medical expertise and official recommendations, play a pivotal role in shaping maternal knowledge of vaccination schedules, benefits, and safety. In addition, friends, relatives, the internet, and social media contribute to their overall landscape of knowledge.

The study showed a highly positive and medically informed attitude, which is in congruence with the observations made by Adefolalu et al. [38] in Nigeria who found a positive attitude among all participated mothers. This was evident through mothers' strong agreement on the benefits of vaccinations, encouraging relatives to vaccinate, and the willingness to add elective vaccines to the national program. Such results underscore the mothers' active role in advocating for broader disease prevention and promoting the dissemination of elective vaccinations in society [39].

The attitude toward pharmacies providing vaccines appears moderately positive in the current study, this can be attributed to the fact that pharmacists in Jordan are not currently regarded as immunizers and till now their role was limited to providing vaccinations to other medical experts [19]. This context is different from previous studies conducted in developed countries like the USA, Ireland, and Canada. Pharmacists in developed countries are considered approved and reliable vaccination providers which boosts vaccination coverage and healthcare financial savings [40–42]. Therefore, it is important to promote pharmacists' role as vaccination providers to spread awareness and a more positive attitude towards vaccination-related advantages.

The study's results reflected a high commitment to medical principles of disease prevention and child health which is consistent with the results of Alshammari et al. [43], which showed that 93% of parents prioritize adherence to vaccination, and Almutairi et al. [25] with a practice score of 80.5%. The consistently high scores across various adherence items showed mothers' dedication to providing their children with vaccinations against VPDs. The commitment to mandatory childhood vaccines and adherence to compulsory vaccination programs demonstrates an awareness of the medical importance of establishing immunity early in life [38]. On the other hand, it could also be a result of the mandatory requirement of the vaccination document for school registration which may encourage the parents to get their children vaccinated.

However, the practice of giving elective vaccines (not included in the national program) is moderate, which indicates a moderate interest in additional protection. The reasons for mothers' non-adherence to elective vaccinations among children's vaccination practices in Jordan illuminate critical factors shaping their decisions. Notably, the satisfaction with the national vaccination program underscores the need for medical professionals to educate mothers about the supplementary advantages of elective vaccines. Lack of awareness about elective vaccines is a significant issue, emphasizing the need for improved medical education efforts to inform mothers about the existence, benefits, and disease-preventing potential of elective vaccines.

Concerns related to high costs highlight the economic aspect of vaccination decisions, showing the importance of medical professionals emphasizing the long-term health and financial advantages of preventive measures. Additionally, addressing fear of side effects through evidence-based information dissemination is essential to counter misconceptions and build trust. Accordingly, facilitating access by providing clear guidance on where to find elective vaccines can bridge information gaps and facilitate informed decision-making for optimal child health protection.

## Conclusions

This study contributes significantly to the understanding of mothers' knowledge, attitudes, and adherence to children's vaccinations in Jordan. Also, identified primary healthcare centers, physicians, and personal networks as key sources of vaccination information. The study provided crucial insights for shaping vaccination campaigns and strategies that promote equitable access to vaccination and enhance informed decision-making. To promote awareness of vaccinations, a hyperlink including a booklet on the types of vaccines in Jordan was provided for respondents who participated in this study and opted-in to this information.

Based on the findings of this study, several recommendations can be suggested. Firstly, developing and implementing tailored health education campaigns that focus on addressing specific knowledge gaps, such as misconceptions about contraindications, side effects and the additional elective vaccines available in Jordan as a boarder vaccination prevention and their added value. Moreover, to highlight the children's most beneficial age groups or special medical conditions such as, chronic illnesses or immunocompromised conditions for targeted elective vaccination recommendations. These campaigns should utilize trusted sources like healthcare professionals and primary healthcare centers to disseminate accurate and evidence-based information. Secondly, providing financial assistance, flexible schedules, and convenient locations to overcome economic barriers and ensure that all children have timely access to vaccinations. Thirdly, establishing an evaluation system to continuously assess the impact of vaccination programs and initiatives. Fourthly, the integration of pharmacists as primary immunization providers in Jordan's healthcare system requires the cooperation of governmental bodies, professionals' associations, and academic institutions. Through an implementation of comprehensive training programs aiming at providing pharmacists with the requisite knowledge and abilities to administer vaccinations efficiently, thus increasing the access to immunization services. Finally, regularly collect data on vaccination coverage, maternal knowledge, and barriers faced to refine strategies and ensure their effectiveness.

Certain limitations of the current study should be acknowledged. This study adopted a cross-sectional design which provides a snapshot of data at a single point in time, while longitudinal or experimental designs are better suited for establishing causality and understanding changes over time. Therefore, future longitudinal studies could show how these concepts evolve over time and in response to changing health communication strategies. The study relied on self-reported surveys, which may be subject to recall bias or social desirability bias. Future research could incorporate structured interviews supported by qualitative data. Moreover, the study's findings were derived from mothers mainly located in the two largest cities in Jordan. Future studies are recommended to encourage participation from other cities in addition to rural areas. Also, the study's findings apply to Jordan's context and may vary when studied in other regions or cultures. Future research could explore similar concepts in different settings or countries to ascertain the generalizability of the findings.

## Acknowledgments

The authors are thankful to the Deanship of Research at Jordan University of Science and Technology for supporting this research.

## Author Contributions

**Conceptualization:** Rand A. Al-Rashdan, Wafa'a Ta'an, Tareq Mukattash, Brett Williams.

**Data curation:** Rand A. Al-Rashdan, Wafa'a Ta'an.

**Formal analysis:** Rand A. Al-Rashdan, Wafa'a Ta'an.

**Investigation:** Wafa'a Ta'an.

**Methodology:** Rand A. Al-Rashdan, Wafa'a Ta'an, Tareq Mukattash, Brett Williams.

**Supervision:** Wafa'a Ta'an, Tareq Mukattash.

**Writing – original draft:** Rand A. Al-Rashdan, Wafa'a Ta'an, Tareq Mukattash, Brett Williams.

**Writing – review & editing:** Rand A. Al-Rashdan, Wafa'a Ta'an, Tareq Mukattash, Brett Williams.

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
