## [Decision Letter · Decision Letter 0]

9 Apr 2024

PONE-D-24-05111Maternal Skepticism Regarding Children's Vaccination in Jordan: Investigating the Knowledge, Attitude, and AdherencePLOS ONE

Dear Dr. Ta'an,

Thank you for submitting your manuscript to PLOS ONE. After careful consideration, we feel that it has merit but does not fully meet PLOS ONE’s publication criteria as it currently stands. Therefore, we invite you to submit a revised version of the manuscript that addresses the points raised during the review process.

We look forward to receiving your revised manuscript.

Kind regards,

Ahmad Haroun Al-Nawafleh, Ph.D, MPA, CI, RN

Academic Editor

PLOS ONE

Journal Requirements:

3. In the online submission form, you indicated that "The data underlying the results presented in the study are available from the corresponding author at wftaan@just.edu.jo upon reasonable request with permission from Jordan University of Science and Technology."

**Additional Editor Comments:**

Dear Authors,

Thank you for this manuscript submission. I am attaching the reviewers comments and I hope you can respond to each point highlighted by both reviewers. Could you please explain the significance of your study in light of the high adherence rate to the mandatory vaccination program in Jordan. Although all the background was built on MOH program adherence, the elective vaccinations in Jordan are very limited, and only the COVID-19 vaccination had an issue.

I look forward to receive your reply in order to consider a final decision on your paper.

Kind regards,

Reviewers' comments:

Reviewer's Responses to Questions

**Comments to the Author**

1. Is the manuscript technically sound, and do the data support the conclusions?

Reviewer #1: Yes

Reviewer #2: No

2. Has the statistical analysis been performed appropriately and rigorously? 

Reviewer #1: Yes

Reviewer #2: No

3. Have the authors made all data underlying the findings in their manuscript fully available?

Reviewer #1: Yes

Reviewer #2: No

4. Is the manuscript presented in an intelligible fashion and written in standard English?

Reviewer #1: No

Reviewer #2: Yes

5. Review Comments to the Author

Reviewer #1: This study provides unique insight into Jordanian vaccination knowledge, attitudes, and practices. The attached file contains some minor revisions that will enhance the clarity and comprehensiveness of the section.

Reviewer #2: dear authors,

kindly check the file attached for specific comments. Overall, the study needs some modifications in the introduction as well as the discussion area, and the results need further correlation to the influencing factors.

thank you ,

6. PLOS authors have the option to publish the peer review history of their article (what does this mean?). If published, this will include your full peer review and any attached files.

If you choose “no”, your identity will remain anonymous, but your review may still be made public.

Reviewer #1: **Yes:**

Reviewer #2: No

---

## [Author Response · Author response to Decision Letter 0]

22 May 2024

Dear respected editor and reviewers, 

We would like to thank the editor and reviewers for the time and effort invested in reviewing our manuscript titled “Maternal Skepticism Regarding Children's Vaccination in Jordan: Investigating the Knowledge, Attitude, and Adherence”. 

We appreciate the thoughtful and constructive feedback and carefully considered each comment and suggestion in preparing this revised manuscript. As you will see in the below responses to the reviewers' comments, we have made the requested revisions to the manuscript to address the concerns raised. 

We hope that the revised manuscript meets the standards of your esteemed journal, and we look forward to your feedback. Once again, we are grateful for the opportunity to resubmit my work to your journal. Please note that all the changes made to the manuscript are in yellow highlight to facilitate the review process. We have also included a point-by-point response to each of the comments made by the reviewer.

---

## [Editor Report · Decision Letter 1]

27 May 2024

Maternal Skepticism Regarding Children's Vaccination in Jordan: Investigating the Knowledge, Attitude, and Adherence

PONE-D-24-05111R1

Dear Dr. Ta'an,

We’re pleased to inform you that your manuscript has been judged scientifically suitable for publication and will be formally accepted for publication once it meets all outstanding technical requirements.

Kind regards,

Ahmad H. Al-Nawafleh, Ph.D, MPA, CI, RN

Academic Editor

PLOS ONE

---

## [Editor Report · Acceptance letter]

5 Jun 2024

PONE-D-24-05111R1 

PLOS ONE

Dear Dr. Ta'an, 

I'm pleased to inform you that your manuscript has been deemed suitable for publication in PLOS ONE. Congratulations! Your manuscript is now being handed over to our production team.

Kind regards, 

on behalf of

Dr. Ahmad H. Al-Nawafleh 

Academic Editor

PLOS ONE